# Effect of Danofloxacin Treatment on the Development of Fluoroquinolone Resistance in *Campylobacter jejuni* in Calves

**DOI:** 10.3390/antibiotics11040531

**Published:** 2022-04-15

**Authors:** Debora Brito Goulart, Ashenafi Feyisa Beyi, Zuowei Wu, Mehmet Cemal Adiguzel, Anastasia Schroeder, Kritika Singh, Changyun Xu, Melda Meral Ocal, Renee Dewell, Grant A. Dewell, Paul J. Plummer, Qijing Zhang, Orhan Sahin

**Affiliations:** 1Department of Veterinary Microbiology and Preventive Medicine, College of Veterinary Medicine, Iowa State University, Ames, IA 50011, USA; dgoulart@iastate.edu (D.B.G.); afbeyi@iastate.edu (A.F.B.); wuzw@iastate.edu (Z.W.); mcemaladiguzel@gmail.com (M.C.A.); schroana@iastate.edu (A.S.); kritika.39@gmail.com (K.S.); xuchangyun1987@gmail.com (C.X.); meralmelda@hotmail.com (M.M.O.); pplummer@iastate.edu (P.J.P.); zhang123@iastate.edu (Q.Z.); 2Center for Food Security and Public Health, College of Veterinary Medicine, Iowa State University, Ames, IA 50011, USA; rdewell@iastate.edu; 3Department of Veterinary Diagnostic and Production Animal Medicine, College of Veterinary Medicine, Iowa State University, Ames, IA 50011, USA; gdewell@iastate.edu; 4National Institute of Antimicrobial Resistance Research and Education, Iowa State University, Ames, IA 50010, USA

**Keywords:** *Campylobacter*, cattle, colonization, rectal feces, bovine respiratory disease (BRD), fluoroquinolone/danofloxacin treatment, antimicrobial resistance, pulsed-field gel electrophoresis, minimum inhibitory concentration

## Abstract

*Campylobacter* is a leading cause of foodborne gastroenteritis. Recent studies have indicated a rise in fluoroquinolone-resistant (FQ-R) *Campylobacter* in cattle, where FQ is used to control bovine respiratory disease (BRD). To assess the effect of danofloxacin treatment on the development of FQ-resistance in *C. jejuni*, 30 commercial calves were divided into Group 1, Group 2, and Group 3 (*n* = 10), and were all inoculated orally with FQ-susceptible (FQ-S) *C. jejuni*; seven days later, Group 3 was challenged with transtracheal *Mannheimia haemolytica*, and one week later, Group 2 and Group 3 were injected subcutaneously with danofloxacin. Rectal feces were collected to determine relative percentages of FQ-R *Campylobacter* via culture. Before oral inoculation with *C. jejuni*, 87% of calves were naturally colonized by FQ-R *C. jejuni*. Two days after the inoculation, FQ-R *C. jejuni* decreased substantially in the majority of calves. Within 24 h of danofloxacin injection, almost all *C. jejuni* populations shifted to an FQ-R phenotype in both FQ-treated groups, which was only transitory, as FQ-S strains became predominant during later periods. Genotyping indicated that the spike seen in FQ-R *C. jejuni* populations following the injection was due mainly to enrichment of preexisting FQ-R *C. jejuni*, rather than development of de novo FQ resistance in susceptible strains. These results provide important insights into the dynamic changes of FQ-resistant *Campylobacter* in cattle in response to FQ treatment.

## 1. Introduction

*Campylobacter* is among the most common causes of foodborne bacterial gastroenteritis globally, and is a significant public health concern, responsible for an estimated 95 million episodes of diarrhea worldwide [1,2,3]. Human *Campylobacter* infections are primarily caused by the consumption of contaminated poultry meat and raw milk [4,5]. Thus, poultry and ruminants serve as the primary reservoirs for this zoonotic organism. Although most patients infected with *Campylobacter* do not require antibiotic treatment, antimicrobial therapy is necessary for severe and systemic infections [6,7]. In these circumstances, macrolides (e.g., erythromycin) and fluoroquinolones (FQs) (e.g., ciprofloxacin) are commonly used for the clinical treatment of human *Campylobacter* infections [6,7,8,9]. However, resistance to both classes of antibiotics—especially to FQs—is increasing in prevalence worldwide [10,11,12], threatening the effectiveness of antimicrobial treatments.

*Campylobacter* is commonly present on cattle farms, and is prevalent in both beef and dairy cattle—usually without causing overt clinical signs [13,14,15]. As *Campylobacter* frequently colonizes the gastrointestinal tracts of cattle, the organism is readily and unavoidably exposed to antibiotics used for the treatment and control of infectious diseases caused by other pathogens, including bovine respiratory disease (BRD). It is important to emphasize that there is no direct association between *Campylobacter* and the development of BRD in cattle. BRD is one of the most widespread endemic diseases in U.S. feedlots, costing the industry about USD 4 billion annually (associated with disease treatment and prevention), along with increased production losses [16,17]. The etiology of BRD is multifactorial, involving multiple viruses and bacteria (primarily *Mannheimia haemolytica*), hosts, and environmental factors [18]. Calves considered to be at high risk of developing BRD are commonly treated metaphylactically with FQs upon feedlot arrival [19,20,21].

Currently, there are two FQ antimicrobials (i.e., enrofloxacin and danofloxacin) approved as injectable solutions for beef cattle and non-lactating dairy cattle in the U.S., and their labels allow for a variety of different doses and treatment schemes [22,23,24]. Both antibiotics have indications for subcutaneous use in both sick animals (therapeutic treatment) and healthy animals (metaphylaxis) that are at high risk of BRD [20,21,25]. Danofloxacin is used as both treatment and metaphylaxis when given subcutaneously at 8 mg/kg of body weight as a one-time injection, or as a multiday treatment therapy when administered subcutaneously at 6 mg/kg, followed by a second dose approximately 48 h later [26,27]. Enrofloxacin is used for both treatment and metaphylaxis when given subcutaneously as a single dose of 7.5 or 12.5 mg/kg of body weight, or as a multiday therapy for the treatment of BRD when administered subcutaneously at 2.5 or 5 mg/kg, followed by additional doses at 24 h, 48 h, and 72 h after the first dose [23,27,28].

There has been a rising trend of fluoroquinolone resistance in ruminant *Campylobacter* over the last two decades in the U.S. Bae et al. found a low rate of ciprofloxacin resistance in *Campylobacter jejuni* (about 5%) from various cattle production types in the Western U.S., even though *Campylobacter coli* isolates from the same study showed a substantially higher resistance rate (about 45%) in 2002 and 2003 [29]. In the mid-2000s, a study of *Campylobacter* isolates from dairy cattle in the Midwest U.S. found that less than 1% of isolates were ciprofloxacin-resistant [30]. In contrast, more recent studies revealed a dramatic increase in the prevalence of FQ-resistant *Campylobacter* in ruminants. A comprehensive study conducted in late 2008 found that a high percentage of *C. jejuni* and *C. coli* (27.3% and 49.2%, respectively) from different cattle populations (including both feedlot cattle and adult cows) were resistant to ciprofloxacin [31]. Importantly, our recent study on *Campylobacter* isolates from 35 feedlots in 5 states in the U.S. revealed an even greater prevalence of ciprofloxacin resistance (35.4% in *C. jejuni* and 74.4% in *C. coli*) during 2012 and 2013 [14].

Since FQ antibiotics are frequently used as therapeutics for BRD in cattle production in the U.S., it is important to assess whether their use influences the development of FQ-resistant *Campylobacter* in the ruminant reservoir, and whether the treatment regimen (i.e., the antibiotic and dose selected) can be managed to reduce the development and prevalence of FQ resistance. We hypothesized that different combinations of FQ drugs (enrofloxacin vs. danofloxacin), the purpose of usage (treatment vs. control), and the condition of the animals (healthy vs. BRD-induced) would show variable effects on the development of FQ-resistant *Campylobacter* in cattle. The main aim of this study was to investigate the effect of a single-dose regimen (the most common form used in U.S. feedlots) of danofloxacin on the development and dynamics of FQ resistance in *C. jejuni* in both healthy and disease-induced calves.

## 2. Results

### 2.1. Natural Campylobacter Colonization Is Common in Calves

The summary of major procedures performed during this study is shown in Table 1.

Fecal culturing showed that the vast majority of the animals (26/30; 87%) were naturally colonized by *C. jejuni* prior to experimental inoculation (days post-inoculation (DPI) −2 and 0 in Figure 1a,c,e). Differential culture plating further indicated the presence of FQ-resistant *C. jejuni* in all of the colonized animals (26/26; 100%) on one or both of the sampling days (DPI −2 and 0 in Figure 1b,d,f). The percentage of FQ-resistant *C. jejuni* colonies relative to the total (susceptible plus resistant) *C. jejuni* population in colonized animals was 51% overall, and was comparable between the groups (63% in Group 3, 47.5% in Group 2, and 42.5% in Group 1; Figure 2a). These data collectively indicated that a significant proportion of the natural *C. jejuni* populations colonizing the calves prior to experimental inoculation were FQ-resistant.

### 2.2. Bovine Respiratory Disease Induction

Given that one of the primary uses of danofloxacin is to treat calves with BRD, we elected to replicate this in one group of calves to determine whether the concurrent disease would affect the development of resistance in *C. jejuni* that may simultaneously be present in the intestine as a commensal organism. The calves in Group 3 were challenged with *M. haemolytica* and monitored for the signs of BRD for the following week using a previously established scoring system [32]. The majority of animals (8/10) in Group 3 developed a BRD-positive score, while only one calf (out of a total of 20) in the two other groups (which did not receive *M. haemolytica*) had a positive score during the one-week period. None of the animals showed any signs of BRD prior to the inoculation. At necropsy, 5/10 calves in Group 3 had typical lung lesions (e.g., consolidation, rough surface, hyperemia), while no significant lesions of such kinds were observed in any of the calves in the two other groups. *M. haemolytica* was cultured from all five of the affected lungs in Group 3, but not from any of the lungs of calves in the other two groups. Based on all of these parameters (clinical scoring, gross pathology, and culture results), BRD induction was considered to be mild–moderate in Group 3, while the two other groups were deemed to have no significant signs of the disease [32,33].

### 2.3. Experimental Inoculation of Calves with FQ-Susceptible C. jejuni Transiently Displaced Preexisting Natural FQ-Resistant C. jejuni Populations

Two days after oral inoculation with a two-strain cocktail of FQ-susceptible *C. jejuni*, all but one animal (29/30; 97%)—including all four animals that were *Campylobacter*-negative prior to inoculation—became colonized by total (susceptible + resistant) *C. jejuni* (DPI 2, Figure 1a,c,e). In contrast, the total number of animals colonized with FQ-resistant *C. jejuni* reduced from the pre-inoculation level of 26/30 (87%) to 20/30 (67%) (DPI 2, Figure 1b,d,f). Similarly, even though the mean colonization levels (CFU/g feces) by total *C. jejuni* increased substantially, the colonization levels by FQ-resistant *C. jejuni* dropped notably in all groups compared with the pre-inoculation levels (Figure 1). In agreement with these decreases in the absolute values of FQ-resistant isolates, the average percentage of FQ-resistant *C. jejuni* isolates relative to the total *C. jejuni* population in colonized animals decreased significantly—particularly in Group 1 and Group 3 (Figure 2a). These observations overall remained similar during the next two sampling points (especially on DPI 5 and, to a lesser extent, DPI 7), despite some variations among the groups (Figure 1). On DPI 14 (fecal samples were collected right before danofloxacin injection in Group 2 and Group 3), both the numbers of calves colonized and the levels of colonization by both the total and FQ-resistant *C. jejuni* were highly comparable to and/or even higher than the pre-inoculation levels (Figure 1). Similarly, the percentages of FQ-resistant *C. jejuni* colonies relative to the total *C. jejuni* population increased gradually, and were somewhat comparable to those found prior to inoculation with FQ-susceptible *C. jejuni* (DPI 14, Figure 2a).

### 2.4. Danofloxacin Treatment Conferred a Transient Fitness Advantage on FQ-Resistant C. jejuni in the Intestines of Calves

Calves in Group 2 and Group 3 were injected subcutaneously with a single dose of danofloxacin two weeks after the oral inoculation with FQ-susceptible *C. jejuni* (i.e., DPI 14; Figure 1 and Figure 2). Twenty-four hours after danofloxacin injection (DPI 15), both the number of animals colonized and the mean colonization levels by total *C. jejuni* population and/or FQ-resistant *C. jejuni* dropped considerably compared with those at pre-injection on DPI 14 (Figure 1). Importantly, the average percentage of FQ-resistant *C. jejuni* relative to the total *C. jejuni* population in colonized animals showed a significant increase in both Group 2 and Group 3 on DPI 15 compared with DPI 14 (Figure 2a). Interestingly, no such dramatic changes were seen in Group 1. During the next two sampling points (DPIs 18 and 21), the number of colonized animals and the mean colonization levels by both the total and FQ-resistant *C. jejuni* populations increased noticeably in Group 2 and Group 3, with some fluctuations seen between the groups (Figure 1). Interestingly, the percentages of FQ-resistant *C. jejuni* isolates relative to the total *C. jejuni* population declined gradually and substantially on DPI 18 and 21 in both Group 2 and Group 3 compared with those seen on DPI 15 (Figure 2a). On DPI 24, almost all of the animals in Group 2 and Group 3 were colonized by quite high levels of both the total and FQ-resistant *C. jejuni* (Figure 1). Importantly, the average percentages of FQ-resistant *C. jejuni* colonies relative to the total *C. jejuni* population decreased sharply in Group 2 and remained low in Group 3 (Figure 2a), which were somewhat comparable to the levels prior to the antibiotic injection (DPI 14). Interestingly, no major dynamic changes were observed in the percentage of FQ-resistant *C. jejuni* in Group 1 on the sampling days beyond DPI 14 (Figure 2a).

### 2.5. Antimicrobial Resistance Profiles of C. jejuni Isolates from Calves

Antibiotic MICs of the isolates (one isolate per positive animal on each sampling day was tested; 262 isolates in total) were determined using the Sensititre panel. Overall, the percentages of the FQ-resistant *C. jejuni* population assessed by the MIC test were comparable to those obtained by the differential plating method throughout the experiment (Figure 2a,b). The MIC values and trends of nalidixic acid (a quinolone antibiotic) were highly similar to those of ciprofloxacin. Additionally, a high level of tetracycline resistance was evident in the *C. jejuni* isolates. For all other antibiotics tested—including azithromycin, erythromycin, florfenicol, tetracycline, gentamicin, and clindamycin—the MIC values were low, and did not appear to fluctuate during the experiment (results not shown).

### 2.6. Danofloxacin Treatment Enriched Preexisting FQ-Resistant C. jejuni Rather Than Inducing De Novo Resistance Development

Genotyping was performed in order to monitor and track the source of FQ resistance in the *C. jejuni* isolates (one isolate per animal on each sampling day was tested; 262 total isolates). PFGE typing generated 16 unique macrorestriction profiles (designated genotypes a–p). Table 2 depicts the occurrence of genotypes at different stages of the experiment, including pre-inoculation, post-inoculation with FQ-susceptible *C. jejuni* (pre-injection with danofloxacin), and post-injection with danofloxacin, along with the ciprofloxacin susceptibility profiles of isolates in Group 2 and Group 3.

Of the 32 total isolates collected pre-inoculation (DPI −2 and 0) from each *Campylobacter*-positive calf in Group 2 and Group 3, the vast majority (*n* = 27) were of the same genotype (a), and were ciprofloxacin-resistant. Following oral inoculation with FQ-susceptible laboratory *C. jejuni* strains (DPI 2–14; pre-injection with danofloxacin), genotype a (including 24 strains—all ciprofloxacin-resistant) remained common, but was predominated by another genotype (e) that contained mainly ciprofloxacin-susceptible (*n* = 24) and several resistant (*n* = 6) strains. Interestingly, neither of the inoculum strains was of genotype e. The remaining 21 isolates were represented by 6 different genotypes, comprising primarily ciprofloxacin-susceptible isolates, including one of the inoculum strains (Table 2). After the subcutaneous danofloxacin injection (DPI 15–24), genotype a regained the predominance (*n* = 31 isolates—all ciprofloxacin-resistant). Almost all of the remaining 36 isolates were ciprofloxacin-susceptible (including the two inoculum strains), and were represented by 10 genotypes.

PFGE genotypes and ciprofloxacin MICs of the *C. jejuni* isolates collected from calves in Group 1 are shown separately in Table 3, as this group did not receive danofloxacin. The results were overall similar to those described for Group 2 and Group 3. Genotype a, comprising all ciprofloxacin-resistant isolates (*n* = 12), was the predominant genotype prior to inoculation with the laboratory strains of *C. jejuni*. Likewise, the diversity of genotypes increased tremendously post-inoculation, with genotype e being the predominant one (*n* = 20 isolates—all but one ciprofloxacin-susceptible), followed by genotype a (*n* = 16—all ciprofloxacin-resistant). The remaining 36 isolates were primarily ciprofloxacin-susceptible (including the two inoculum strains), and were represented by 9 genotypes (Table 3).

## 3. Discussion

Antimicrobial use in food animal production has sparked a contentious debate in recent years due to concerns about the transmission of resistant bacteria from animal food products to humans [34]. To assess the risk of antimicrobial usage in agriculture, we investigated the effect of danofloxacin—an FQ antibiotic—on the development of FQ resistance in *C. jejuni* in both healthy and disease-induced (BRD) calves. To the best of our knowledge, this is the first study in which calves were orally inoculated with *C. jejuni* and subsequently challenged with *M. haemolytica* and treated with an FQ antibiotic. A key finding of this study was that a single-dose subcutaneous danofloxacin treatment in calves caused a rapid and sharp, yet transient, enrichment of preexisting FQ-resistant *C. jejuni* populations in the intestine, but did not appear to result in any measurable level of de novo FQ resistance development from the inoculated FQ-susceptible *C. jejuni*.

The majority of the calves were naturally infected with *C. jejuni* on the day of arrival in the present study. This was unexpected, but was not entirely surprising, as *Campylobacter* is common (up to 80–90% farm- and within-herd-level prevalence rates) in both dairy and feedlot cattle in various geographic regions of the world, including the U.S. [13,15,35,36,37,38,39,40]. Horizontal transmission from environmental sources (e.g., livestock, poultry and other fowl, wildlife, the immediate farm environment—including manure—personnel, etc.) is commonly involved in introducing *Campylobacter* to cattle farms [40,41]. The naturally colonized calves used in this study yielded an average of 3.8 log_10_ CFU/g feces of *C. jejuni* prior to inoculation with laboratory strains, which is comparable to that found in feces of dairy calves at four months of age (3.7 to 4.4 log_10_ CFU/g feces) [42], but lower than that found in broiler chicken feces (up to 9.0 log_10_ CFU/g feces) [43]. Notably, the calves had no history of known exposure to FQ antibiotics according to their treatment records and the farm veterinarian. Although the exact source of FQ-resistant *C. jejuni* in the studied calves is not known, it is possible that it was acquired directly from the farm environment. While commencing the study with most of the calves infected with FQ-resistant *C. jejuni* was a shortcoming, finding *Campylobacter*-free calves was difficult for us, despite repeated testing of candidate farms; thus, it was decided to proceed with the study as planned. To overcome this limitation, we performed PFGE genotyping of the isolates to differentiate the de novo resistance development in the inoculated susceptible isolates from the selection of preexisting FQ-resistant strains. Several investigations of the development of antimicrobial resistance have also faced similar issues, with animals already colonized by resistant strains at the beginning of the study [44,45,46,47,48,49,50]. It should be reemphasized that prevalence of *Campylobacter* in general and FQ resistance in particular in cattle has been on the rise worldwide, reaching as high 100% at the farm level and 46–72% at the within-herd level in U.S. feedlots in recent years [14], making it a challenging task to find *Campylobacter*-negative animals from commercial sources for research purposes.

Soon after oral inoculation with FQ-susceptible laboratory *C. jejuni* strains, both the absolute levels and the relative percentages of FQ-resistant *C. jejuni* showed a sharp decrease, but gradually increased during the following days, and returned to levels that were fairly comparable to the pre-inoculation values in all groups by DPI 14 (Figure 2a). These findings indicated that even though FQ-susceptible *C. jejuni* could displace preexisting natural FQ-resistant *C. jejuni* populations in the intestines of calves soon after experimental inoculation with the susceptible strains, this was only transient, and the FQ-resistant population was able to repopulate to close to the pre-inoculation level within two weeks, with no antibiotic selection pressure present, regardless of BRD status. It is important to point out that the more rapid recovery of FQ-resistant *C. jejuni* in Group 3 (BRD-induced group) occurred prior to induction of the disease; therefore, it is not a likely effect of the *M. haemolytica* challenge to this group. Although a direct comparison is not feasible (due to the differences in the experimental designs between the studies), this observation (i.e., the overall fitness of FQ-resistant *C. jejuni*) is somewhat similar to that found in a previous study, in which a remarkable fitness advantage of FQ-resistant *C. jejuni* over FQ-susceptible isolates was demonstrated using isogenic strains in the absence of antibiotic selection pressure following in vivo co-inoculation experiments [51].

It is interesting to note that even though the subcutaneous danofloxacin injection performed on DPI 14 substantially reduced both the number of colonized calves and the level of colonization by either total or FQ-resistant *C. jejuni* in Group 3 (BRD-induced group) a day after the injection, such dramatic changes were not observed in Group 2—especially not in the level of colonization by FQ-resistant *C. jejuni* (Figure 1). Whether or not this finding was truly associated with the BRD status of the calves cannot be definitively deduced based on the available data in this study. Interestingly, the overall kinetics of danofloxacin in both plasma (peaked at 2–4 h after injection) and feces (peaked at 8–12 h) in both groups were comparable, even though the drug concentrations were noticeably higher in Group 2 during the peak period [52]. However, this sharp decrease in the colonization levels was only transient (less than 4 days after the injection), and they quickly returned to levels comparable to the pre-injection values in the following days. Importantly, and regardless of the aforementioned differences in the colonization levels between the two groups in response to the antibiotic exposure, the average percentage of FQ-resistant *C. jejuni* relative to the total *C. jejuni* population underwent a highly significant and sharp increase as early as a day after the danofloxacin injection in both groups (Figure 2a). This predominance by FQ-resistant *C. jejuni* gradually decreased, and the FQ-resistant population in both groups constituted less than half of the total population on day 10 after the antibiotic injection. Collectively, these findings indicate that a single-dose subcutaneous danofloxacin treatment of calves—as employed in the current study—confers a rapid yet transient fitness advantage to FQ-resistant *C. jejuni* over FQ-susceptible *C. jejuni* in the intestines of cattle, with only marginal differences being observed with regard to the BRD status of the animals.

The main objective of this study was to determine the effect of danofloxacin treatment on the development of FQ resistance in *C. jejuni* in cattle. However, even though the danofloxacin injection resulted in a swift and striking increase in the relative percentage of FQ-resistant *C. jejuni* as described above, the underlying mechanism(s) for this observation could not be immediately ascertained with the available data. This was due to the fact that the vast majority of calves were already colonized with FQ-resistant *C. jejuni* prior to experimental inoculation with FQ-susceptible strains. Therefore, PFGE genotyping of the isolates (one isolate per animal on each sampling day) was performed to determine whether the spike seen in FQ resistance following the danofloxacin injection was due to the development of de novo resistance from previously susceptible strains, or due to the selection of preexisting FQ-resistant *C. jejuni* strains (or both). First and foremost, a close look into the genotyping data (Table 2) indicated that there was a total of 46 ciprofloxacin-susceptible isolates represented by 7 different genotypes prior to danofloxacin injection, and only one of these genotypes had a single ciprofloxacin-resistant isolate detected post-antibiotic-injection. Additionally, it is worthwhile to note that the danofloxacin injection was associated with an increase in genotype diversity, with newly detected genotypes being mostly FQ-susceptible. Lastly, almost all of the FQ-resistant isolates (31/33, 93.9%) detected post-antibiotic-injection were of the same genotype (i.e., genotype a), which was also the predominant (24/33, 72.7%) FQ-resistant genotype detected pre-antibiotic-injection. In other words, the resistant *C. jejuni* isolates recovered after the danofloxacin injection most likely represented the reestablishment of the original resistant strains that survived the antibiotic treatment. Collectively, these observations suggest that the single-dose danofloxacin treatment selectively enriched certain preexisting FQ-resistant *C. jejuni* genotypes without having a broad effect on other *C. jejuni* populations, and that it did not appear to result in the development of de novo FQ resistance in FQ-susceptible *C. jejuni* strains in the intestines of the cattle. Interestingly, comparable findings were observed in a recent field study [49], in which feedlot cattle considered to be at risk of BRD development were subjected to fluoroquinolone metaphylaxis (i.e., a single subcutaneous enrofloxacin injection) in a randomized trial. Similar to our study, the majority of *Campylobacter* spp. recovered from the rectal feces of animals prior to the treatment were FQ-resistant and, importantly, the metaphylaxis had no significant effect on the fecal prevalence of FQ-resistant *Campylobacter* during the 4-week-long trial [49]. These observations are in stark contrast to the findings in chicken hosts, in which fluoroquinolone treatment (enrofloxacin, sarafloxacin, or difloxacin; typically given in drinking water for 5 days) results in rapid and sustained development of de novo FQ resistance in preexisting FQ-susceptible *C. jejuni* strains in the intestine under both experimental and commercial settings [46,51,53,54,55]. Limited data available in swine also suggest rapid emergence of FQ-resistant *Campylobacter* from preexisting FQ-susceptible *Campylobacter* in the intestine following multidose administration of enrofloxacin (intramuscular) or norfloxacin (oral) in a laboratory setting [56]. The exact reason(s) for these differences remains to be elucidated, but it is possible that the variations in the treatment regimens and/or the high *Campylobacter* load may provide a more permissive environment for the emergence and persistence of FQ-resistant *Campylobacter* in chickens (and perhaps swine) than in cattle under the antibiotic selection pressure.

Even though the main purpose of PFGE analysis was to determine the sources and mechanisms of FQ resistance in *C. jejuni* in our study, it also provided insights into the genetic diversity of the isolates acquired throughout the study (one isolate per animal on each sampling day; 262 total). In total, 16 unique macrorestriction profiles (including the two inoculum genotypes) were identified, indicating a moderate–high level of genetic heterogeneity among the *C. jejuni* populations tested in the studied calves (Table 2 and Table 3). Overall high levels of genetic diversity of *C. jejuni* isolates in cattle have previously been reported from the U.S. in our recent study [5], as well as from other parts of the world [57,58,59,60]. Notably, distinct macrorestriction profiles were identified from the same calves at different sampling times, while some profiles were detected from different calves on multiple occasions throughout the study (partially shown in Table 2 and Table 3). It was also obvious that certain genotypes (i.e., primarily genotype a, followed by genotype e, and genotype i to a lesser extent) were predominant, and represented the vast majority of *C. jejuni* isolates examined in our study; similar findings were also reported by other investigators [58,59,60]. MLST analysis of the representative isolates of the predominant PFGE patterns showed that genotype a was of ST-982, genotype e was of ST-929, and genotype i was of ST-61. ST-982 has previously been found in ruminants and the environment in the U.S. [14], and frequently recovered from cattle and humans, emphasizing the association of unpasteurized milk consumption with human campylobacteriosis [61,62]. ST-61 has also been previously found in bovines, suggesting that this clone may represent a *C. jejuni* genotype adapted to cattle [63,64].

The present study has some limitations. First, as mentioned earlier, the vast majority of calves procured were already colonized with FQ-resistant *C. jejuni* prior to the start of the experiment. It would have been ideal if the calves had not harbored any *Campylobacter*, or if the *Campylobacter* isolates they had were all FQ-susceptible, for the purpose of this study. Second, only a single *C. jejuni* isolate per animal on each sampling day was selected for further characterization via MIC and PFGE; thus, it is possible that the less common phenotypic and genotypic traits of interest may not have been captured. Finally, BRD induction by *M. haemolytica* challenge was only at mild-to-moderate levels; therefore, it is possible that the true contribution of this condition to the development of FQ resistance was not fully addressed.

## 4. Materials and Methods

### 4.1. Animals and Study Design

Thirty dairy calves (4 females and 26 males) with predominantly Holstein genetics were obtained from a local dairy farm in Iowa in July 2018. Animals were approximately two months old at the time of procurement, and were between 54 and 93 kg in body weight. To be eligible for the study, calves were required to have had no history of antibiotic exposure and have no overt clinical signs of illness at arrival. Iowa State University (ISU) veterinarians visually checked calves as they arrived at the animal facility for symptoms of illness, such as lameness, nasal discharge, dyspnea, obtundation, ophthalmic defects, bloating, and diarrhea. During the study, none of these calves developed any severe health issues that necessitated the use of additional antibiotics. Calves were weighed after a visual inspection and then randomly assigned a unique identification number and group-housed in three different groups (*n* = 10 calves per group): (1) Group 1—oral inoculation with *C. jejuni* only and no additional treatment; (2) Group 3—oral inoculation with *C. jejuni*, followed seven days later by transtracheal inoculation with *Mannheimia haemolytica*, and then followed seven days later by subcutaneous administration of a single dose of danofloxacin; and (3) Group 2—oral inoculation with *C. jejuni*, followed 14 days later by subcutaneous administration of a single dose of danofloxacin (Table 1). Relevant information on the bacterial isolates used in the challenge experiments is presented in Appendix A. Calves were fed mixed grass hay and a premixed calf starter (Heartland Co-op, Des Moines, IA, USA); water was given ad libitum throughout the study. Animals were housed in the Livestock Infectious Disease Isolation Facility (LIDIF) at ISU in biosafety level 2 containment for 28 days. All animals were maintained and handled under the protocols and procedures approved prior to the start of the study by the Institutional Animal Care and Use Committee (IACUC-18-372) at ISU.

### 4.2. Oral Inoculation with Fluoroquinolone-Susceptible C. jejuni

After a four-day acclimatization period at the ISU animal facility, the calves were orally inoculated via an esophageal tube with a mixture of two different FQ-susceptible *C. jejuni* strains (ciprofloxacin MIC = 0.125 μg/mL). These strains (IA-6-FC-30 and MO-2-FC-25) were of different PFGE/MLST subtypes, and were isolated in 2013 from the feces of cattle derived from feedlot herds located in Iowa and Missouri, respectively [65]. Frozen glycerol stocks of each strain were streaked on Mueller–Hinton (MH) agar (BD, Basingstoke, United Kingdom) plates and incubated at 42 °C for about 48 h under microaerobic conditions (10% CO_2_, 5% O_2_, 85% N_2_). Fresh, logarithmic phase culture of each strain was obtained after a sub-passage on fresh MH agar plates for approximately 20 h of incubation under the same conditions as described above. After the incubation, bacterial growth on agar plates was harvested in sterile PBS, and the OD_600_ was adjusted to 1.0. A mixture containing equal volumes of each OD-adjusted strain was prepared, which had a final concentration of ~3 × 10^9^ CFU/mL, as determined by viable plate counts. Each calf was orally inoculated with 60 mL of the final suspension directly into the rumen using an esophageal tube. Interestingly, the two strains were determined to be highly motile on semi-solid agar using methods described elsewhere [66] prior to being used in this study.

### 4.3. Inoculation with Mannheimia haemolytica

One week after the inoculation with the FQ-susceptible *C. jejuni* cocktail, the calves in Group 3 were inoculated with *M. haemolytica* by transtracheal injection using a catheter, as described elsewhere [67], to induce BRD. While the intratracheal respiratory disease challenge model originally described by Gibbs et al. [68] is commonly used in cattle, a number of other techniques are reported in the recent literature, including catheter delivery and nebulization. The continued use of multiple methods suggests that there is not a universal benefit of the Gibbs method compared to others. In a pilot study, we observed somewhat variable induction of respiratory disease with *M. haemolytica* using the Gibbs model, and thus decided to use another challenge model (i.e., transtracheal injection using a catheter), as described elsewhere [67]. The *M. haemolytica* strain was originally isolated from the lungs of a dead calf diagnosed with bacterial pneumonia at the Veterinary Diagnostic Laboratory (VDL) of ISU. The inoculum was prepared as described previously [66]. Briefly, the isolate was recovered on MH agar from frozen glycerol stock at −80 °C, and fresh culture was prepared by sub-passaging on new MH agar plates for overnight incubation at 37 °C. Cells were harvested in sterile saline, centrifuged at 3000× *g* for 20 min, and the pellet was suspended in fresh saline to obtain a suspension with an OD_600_ of 2.0 (~1.0 × 10^9^ CFU/mL). Each calf was inoculated with 10 mL of this suspension via the transtracheal route. Routine health observations to check for BRD symptoms—including fever, depression, ocular and nasal discharges, ear droop or head tilting, cough, and changes in respiration, eating, and ambulation—were monitored and recorded for one week after the inoculation. The calves were categorized as BRD-positive or BRD-negative based on a scoring system as per the clinical parameters described elsewhere [32].

### 4.4. Danofloxacin Injection

All of the calves in Group 2 and Group 3 were subcutaneously (sc) injected with a single dose of danofloxacin (8 mg/kg body weight, ADVOCIN™, danofloxacin mesylate (Zoetis) in the neck 14 days after the *C. jejuni* inoculation (i.e., 7 days after the *M. haemolytica* inoculation in Group 3).

### 4.5. Fecal Sample Collection

Individual fecal samples (approximately 20 g) were collected directly from the rectum of each animal at different intervals before and after oral inoculation with *C. jejuni* until the end of experiment at necropsy. Samples were collected using separate latex gloves and placed in 50 mL sterile centrifuge tubes labeled with the animal ID and sampling date, and were kept on ice in an insulated foam container. The fecal samples were processed for culture in the laboratory within a few hours of the collection. The collection times correspond to −2, 0, 2, 5, 7, 14, 15, 18, 21, and 24 days post-inoculation; day 0 refers to the day of oral inoculation with the *C. jejuni* strains. It should be noted that the samples on day 0 were obtained prior to the experimental inoculation with *C. jejuni*.

### 4.6. Bacterial Isolation and Identification

For each sample, approximately 3 g of feces was placed into a Ziploc bag with 27 mL of MH broth. Fecal samples were then serially diluted (10-fold) with MH broth. An inoculum of 100 μL from multiple dilutions was streaked onto MH agar plates containing *Campylobacter* growth supplement (SR084E; Oxoid, Basingstoke, UK) and Preston *Campylobacter*-selective supplement (SR117E; Oxoid), and onto MH agar plates containing the same supplements plus ciprofloxacin (4 μg/mL), for counting (CFU/mL) total *Campylobacter* and FQ-resistant *Campylobacter,* respectively. Plates were incubated at 42 °C for 48 h under microaerobic conditions. For further characterization, two *Campylobacter*-like colonies from each animal on each sampling day were picked from the selective MH agar plates without ciprofloxacin, streaked onto new plain MH agar plates, and incubated at 42 °C for 24 h under microaerobic conditions. Pure cultures were collected in MH broth with 30% glycerol and stored at −80 °C until further use.

Enrichment culture was also carried out on fecal samples collected prior to inoculation with *C. jejuni* in order to ensure the *Campylobacter* status of the calves, as this method is more sensitive than direct culture when the numbers of *Campylobacter* in cattle feces are low [69]. For this purpose, approximately 1 g of feces was placed in a 50 mL tube (with a ventilated screw-cap) containing 15 mL of MH broth with the same growth and *Campylobacter*-selective supplements as above, and incubated at 42 °C for 48 h under microaerobic conditions. A small aliquot from the enrichment culture was streaked onto MH agar containing the same growth and selective supplements, and onto MH agar with the same supplements plus ciprofloxacin, and incubated under microaerobic conditions for 48 h at 42 °C. Plates were observed for growth, and two well-isolated *Campylobacter*-like colonies per sample were re-streaked onto plain MH agar plates to obtain pure cultures from samples that did not yield any *Campylobacter*-suspect growth from the direct culture. Confirmation and species-level identification of all of the saved isolates was determined by MALDI-TOF mass spectrometry, following the manufacturer’s (Bruker Daltonik, Billerica, MA, USA) instructions and standard operating procedures at the ISU VDL.

### 4.7. Pulsed-Field Gel Electrophoresis

Pulsed-field gel electrophoresis (PFGE) analysis of *C. jejuni* isolates using the *Sma*I restriction enzyme was performed as described elsewhere [70]. Briefly, freshly prepared cultures of *C. jejuni* were embedded in 1% SeaKem Gold agarose (Fisher Scientific, Fair Lawn, NJ, USA) and lysed with proteinase K (0.5 mg/mL) for 30 min at 50 °C in a shaking water bath. The gel plugs were digested with *Sma*I for 4 h at 37 °C. Digested plugs were embedded into 1% agarose and separated by electrophoresis in 0.5× Tris-borate-EDTA (TBE) buffer at 14 °C for 18 h using a Chef Mapper electrophoresis system (Bio-Rad, Hercules, CA, USA). The gels were stained with ethidium bromide for 30 min and then photographed using the digital imager ChemiImager™ 5500 (Alpha Innotech, Santa Clara, CA, USA). The PFGE patterns were analyzed with GelCompar II v.6.5 (Applied Maths, Kortrijk, Belgium). Restriction fragments were identified visually, and the PFGE patterns were normalized by interpolation to the nearest reference line. The molecular sizes of the fragments were calculated based on the fragments of the Lambda DNA ladder (Bio-Rad) that was used as the molecular size marker. Only fragments in the size range 48.5–1000 kb were analyzed, as smaller fragments were not consistently resolved. An optimization of 0.5% and a position tolerance of 1.5% were applied [71]. Dice similarity coefficients were calculated based on a pairwise comparison of the PFGE profiles of the isolates [72,73]. The coefficients matrix was used to generate dendrograms based on the unweighted pair group method using arithmetic averages (UPGMA) [72,73]. Isolates were considered to be closely related genotypes if their PFGE profiles were clustered together at ≥0.90, as determined by the GelCompar II analysis [72].

### 4.8. Multilocus Sequence Typing

Representative isolates of the PFGE clusters/genotypes were also analyzed by multilocus sequence typing (MLST), following the methods described on the *Campylobacter jejuni*/*coli* PubMLST website (https://pubmlst.org/organisms/campylobacter-jejunicoli, accessed on 25 October 2021) based on the seven housekeeping genes (*aspA*, *glnA*, *glyA*, *pgm*, *tkt*, *uncA*, and *gltA*) [74]. PCR products were purified using the QIAquick^®^ PCR purification kit (QIAGEN, Hilden, Germany) and sequenced at the DNA Core Facility at ISU using an Applied Biosystems 3730 xl DNA Analyzer. Allelic numbers, allelic profiles, and sequence types (STs) were assigned using the different modules of the *Campylobacter jejuni*/*coli* typing database on the PubMLST site.

### 4.9. Minimum Inhibitory Concentration

The minimum inhibitory concentrations (MIC) of eight antibiotics for *C. jejuni* isolates were determined via broth microdilution tests using commercially available Sensititre *Campylobacter* plates (CAMPY2, Thermo Fisher Scientific, Waltham, MA, USA), following the guidelines set out by the Clinical and Laboratory Standards Institute (CLSI) and the National Antimicrobial Resistance Monitoring System for Enteric Bacteria (NARMS) [75,76]. The antimicrobials included were ciprofloxacin (CIP), nalidixic acid (NAL), azithromycin (AZI), erythromycin (ERY), tetracycline (TET), florfenicol (FFC), clindamycin (CLI), and gentamicin (GEN). As a quality control organism, *C. jejuni* ATCC 33560 was used. The lowest antimicrobial concentration at which no visible bacterial growth developed was recorded as the MIC value for each isolate. The CLSI interpretive criteria (µg/mL) for resistance were used as follows: ciprofloxacin (≥4), erythromycin (≥32), and tetracycline (≥16) [75]. As there are no CLSI breakpoints established for other antibiotics, clinical resistance breakpoints (µg/mL) previously used by NARMS were employed as follows: azithromycin (≥8), clindamycin (≥8), florfenicol (≥16), gentamicin (≥8), and nalidixic acid (≥32) [76].

### 4.10. Necropsy

Ten days after the danofloxacin injection, calves were euthanized using a penetrating captive bolt gun. A portion of the lung with hepatization (e.g., hyperemia, rough surface) and consolidation was collected for *Mannheimia* culture. Each lung sample was placed in sterile Petri dish and transferred to the laboratory within an hour of collection. The surface of the lung was stabbed with a sterile cotton swab and the swab was streaked on blood agar, and the plates were incubated aerobically at 37 °C for 24 h. Suspect *M. haemolytica* colonies (based mainly on the presence of hemolysis) were re-streaked onto a new blood agar plate and incubated in the same way as above to obtain pure cultures. The isolates were preserved in MH broth with 30% glycerol at −80 °C until final identification by MALDI-TOF MS.

### 4.11. Statistical Analysis

Significant differences between groups in the percentage of colonized calves with total *Campylobacter* (FQ resistant and sensitive) and FQ-resistant *Campylobacter* at each sampling point were determined using Fisher’s exact test. One-way analysis of variance (ANOVA) followed by Tukey’s post hoc test was used to calculate the significant differences in the colonization levels (log-transformed) of total and FQ-resistant *Campylobacter* at each sampling point between groups and within each group. A *p*-value ≤ 0.05 was considered significant. The data were analyzed using GraphPad software (Prism, San Diego, CA, USA).

## 5. Conclusions

The study design employed here (with a relatively large number of animals in each of the three groups, a repeated sampling scheme, and rigorous phenotypic and genotypic characterization of the isolates) allowed us to conclude that a single-dose subcutaneous danofloxacin treatment does not appear to result in the emergence of de novo FQ resistance in FQ-susceptible *C. jejuni* in the intestines of cattle. However, our findings also indicated that FQ treatment provides a transient selection force for preexisting FQ-resistant *Campylobacter* in cattle, which may contribute to the persistence of FQ resistance on cattle farms. In future studies, it would be highly valuable to evaluate the effects of different treatment schemes (e.g., multidose therapy) and FQ antibiotics (e.g., enrofloxacin) approved for use against BRD in cattle on the development of FQ resistance in *Campylobacter* before well-informed decisions can be drawn.

## Figures and Tables

**Figure 1 antibiotics-11-00531-f001:**
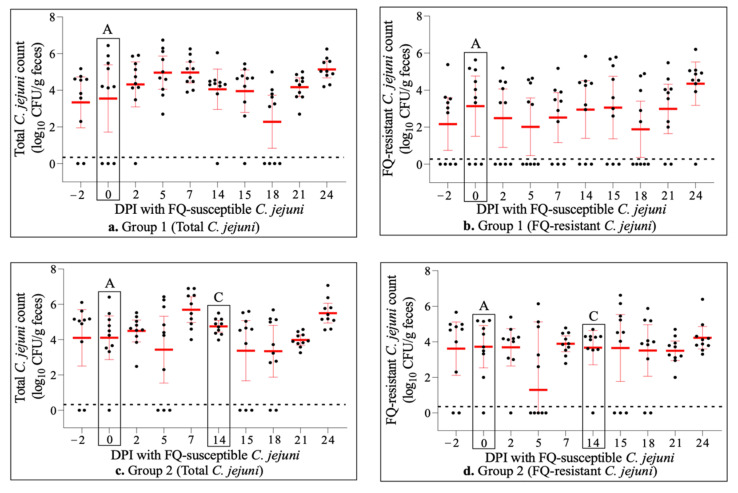
Dot plot graphs showing the colonization level (log_10_ CFU/g feces) by total (susceptible and resistant) *C. jejuni* population (**a**,**c**,**e**) and FQ-resistant *C. jejuni* population (**b**,**d**,**f**) in three groups (Group 1, Group 2, and Group 3) of calves. The letter A denotes the time when all animals were orally inoculated (after fecal samples were collected for culture on DPI 0) with FQ-susceptible *C. jejuni*. The letter B indicates when the calves in Group 3 were challenged (after fecal samples were collected for culture on DPI 7) with *M. haemolytica*. The letter C represents subcutaneous injection (after fecal samples were collected for culture on DPI 14) with danofloxacin in Group 2 and Group 3. Each dot represents the colonization level in a single calf; the mean colonization levels are indicated by horizontal red bars, and vertical red lines represent 95% confidence intervals. The detection limit of the culture was ~100 CFU/g of feces (shown as dotted black lines over the x-axis). DPI: days post-inoculation.

**Figure 2 antibiotics-11-00531-f002:**
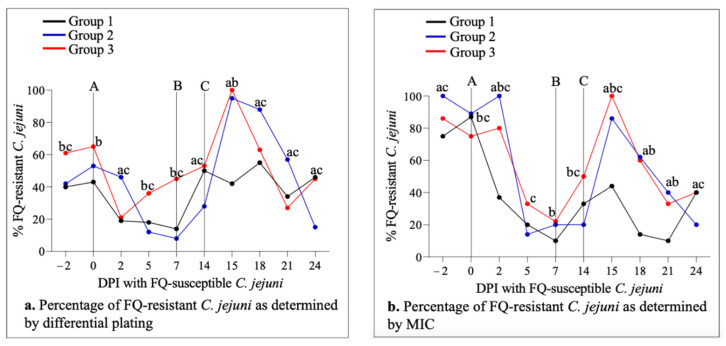
Percentages of FQ-resistant *C. jejuni* colonies (in colonized animals) as determined by differential plating (**a**) and MIC (**b**). Group 1, Group 2, and Group 3 are represented by the black, blue, and red lines, respectively. The letter A represents the time when all calves were orally inoculated (after fecal samples were collected for culture on DPI 0) with FQ-susceptible *C. jejuni*. The letter B indicates when calves in Group 3 were challenged (after fecal samples were collected for culture on DPI 7) with *M. haemolytica*. The letter C represents subcutaneous injection with danofloxacin (after fecal samples were collected for culture on DPI 14) of animals in Group 2 and Group 3. Lowercase letters indicate significant differences (*p* < 0.05) between groups (a: Group 1 vs. Group 2; b: Group 1 vs. Group 3; c: Group 2 vs. Group 3).

**Table 1 antibiotics-11-00531-t001:** Summary of the main experimental procedures performed in the present study.

Group	Inoculation with FQ-S *C. jejuni* *	Challenge with *M. haemolytica* ^#^	Danofloxacin Injection ^§^
1	Yes	No	No
2	Yes	No	Yes
3	Yes	Yes	Yes

* All of the calves were orally inoculated with FQ-S *C. jejuni* on day 0 (days post-inoculation (DPI) 0) after a four-day acclimatization period. ^#^ One week after this inoculation, the calves in Group 3 were given *M. haemolytica* via the transtracheal route (DPI 7). ^§^ Calves in Group 2 and Group 3 received a subcutaneous danofloxacin injection 14 days after the *C. jejuni* inoculation (DPI 14). Fecal sampling was performed throughout the study, including the acclimatization period. Necropsy was performed on DPI 24.

**Table 2 antibiotics-11-00531-t002:** PFGE genotypes and ciprofloxacin susceptibility of *C. jejuni* isolates from calves in Group 2 and Group 3 obtained throughout the study ^#^.

Pre-Inoculation (DPI −2 and 0), *n* = 32			Post-Inoculation (DPI 2–14), *n* = 75			Post-Injection (DPI 15–24), *n* = 67		
Genotype *	CIP ^§^	MIC *	Genotype	CIP	MIC	Genotype	CIP	MIC
a (27)	R	8 (25); 16 (2)	a (24)	R	8 (22); 16 (2)	a (31)	R	8 (30); 16 (1)
b (3)	S	0.06 (3)	b (1)	S	0.06 (1)	b (3)	S/R	0.06 (2); 8 (1)
c (1)	R	8 (1)	c (1)	R	8 (1)	c (0)	---	---
d (1)	S	0.12 (1)	d (0)	---	---	d (0)	---	---
e (0)	---	---	e (30)	S/R	0.06 (3); 0.12 (19); 0.25 (1); 2 (1); 4 (3); 8 (3)	e (7)	S	0.12 (4); 0.06 (3)
f (0)	---	---	f (3)	S	0.12 (3)	f (1)	S	0.12 (1)
g (0)	---	---	g (0)	---	---	g (1)	S	0.12 (1)
h (0)	---	---	h (0)	---	---	h (2)	S/R	0.12 (1); 4 (1)
i (0)	---	---	i (12)	S/R	0.06 (1); 0.12 (9); 8 (2)	i (7)	S	0.12 (7)
j (0)	---	---	j (1)	S	0.12 (1)	j (4)	S	0.12 (4)
k (0)	---	---	k (0)	---	---	k (1)	S	0.12 (1)
l (0) ^¶^	---	---	l (3)	S	0.12 (3)	l (9)	S	0.06 (2); 0.25 (2); 0.12 (5)
m (0) ^¶^	---	---	m (0)	---	---	m (1)	S	0.06 (1)

^#^ DPI −2 and 0 include isolates prior to inoculation with laboratory strains of FQ-susceptible *C. jejuni.* DPI 2–14 represent isolates obtained between post-*C. jejuni*-inoculation and pre-danofloxacin-injection. DPI 15–24 comprise isolates collected post-danofloxacin-injection; “*n*” denotes the number of isolates tested at each period. * Each unique genotype (macrorestriction pattern) is assigned to a different alphabetical letter. Numbers in parentheses indicate the number of isolates with that particular genotype or ciprofloxacin MIC. ^§^ Ciprofloxacin susceptibility phenotype; R denotes resistant (MIC ≥ 4), S denotes susceptible (MIC ≤ 2). ^¶^ Genotypes of the strains used as inoculum.

**Table 3 antibiotics-11-00531-t003:** PFGE genotypes and ciprofloxacin susceptibility of *C. jejuni* isolates from calves in Group 1 obtained throughout the study ^#^.

Pre-Inoculation (DPI −2 and 0), *n* = 16			Post-Inoculation (DPI 2–24), *n* = 72		
Genotype *	CIP ^§^	MIC *	Genotype	CIP	MIC
a (12)	R	4 (1); 8 (10); 16 (1)	a (16)	R	8 (16)
b (4)	S/R	0.06 (2); 0.12 (1); 4 (1)	b (5)	S	0.06 (4); 0.12 (1)
e (0)	---	---	e (20)	S/R	0.12 (16); 0.25 (1); 0.6 (2); 4 (1)
f (0)	---	---	f (4)	S/R	0.12 (3); 8 (1)
i (0)	---	---	i (8)	S	0.06 (1); 0.12 (6); 0.25 (1)
j (0)	---	---	j (5)	S/R	0.12 (4); 4 (1)
l (0) ^¶^	---	---	l (3)	S	0.12 (3)
m (0) ^¶^	---	---	m (4)	S	0.12 (4)
n (0)	---	---	n (4)	S/R	0.12 (3); 8 (1)
o (0)	---	---	o (2)	S	0.12 (2)
p (0)	---	---	p (1)	S	0.12 (1)

^#^ DPI −2 and 0 include isolates prior to inoculation with laboratory strains of FQ-susceptible *C. jejuni.* DPI 2–24 represent isolates obtained between post-*C. jejuni*-inoculation and necropsy (danofloxacin injection was not performed); “*n*” denotes the number of isolates tested at each period. * Each unique genotype (macrorestriction pattern) is assigned to a different alphabetical letter (the same naming convention as in Table 2 was used). Numbers in parentheses indicate the number of isolates with that particular genotype or ciprofloxacin MIC. ^§^ Ciprofloxacin susceptibility phenotype; R denotes resistant (MIC ≥ 4), S denotes susceptible (MIC ≤ 2). ^¶^ Genotypes of the strains used as inoculum.

## Data Availability

Data are available upon request.

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
