# Peer review of "Effect of Danofloxacin Treatment on the Development of Fluoroquinolone Resistance in *Campylobacter jejuni* in Calves"

_antibiotics, 2022, doi:10.3390/antibiotics11040531_

Round 1
Reviewer 1 Report
The study entitled “Subcutaneous Danofloxacin Treatment Transiently Enriches Preexisting Fluoroquinolone Resistant Campylobacter jejuni in Calves” concerns the development of antibiotic resistant strains of C. jejuni in bovine reservoirs. It is an interesting work, but some changes should be made for better understanding of described issues.
- I don’t like a title, please consider a title that is more relevant
- Please try to prepare the introduction in a more concise form with precise aims setting.
- Please stress the relation between jejuni and BRD, this in unclear in present form
- Generally, the introduction should be shorter and more clear.
- Please indicate the main aim of the study.
- In the Results part please try to focus on the most relevant results, all results are shown on tables and figures, in my opinion it is not necessary to describe them such
- The part from 612 to 693 is the replication of results, please discuss your results with previous findings and shorten this part to the key results description and discussion.
- Please prepare a short conclusion from the study
- I have some doubts about the nomenclature of experimental groups, the terms “control group” and “healthy group” are confusing for me. The term “Control group” suggests animals without treatment, whereas they were inoculated with C. jejuni. Control group actually doesn’t exist in this study. “Healthy group”, in turn, represents calves inoculated with C. jejuni and treated with danofluxacin. Please consider change of these names.
- Did the authors receive the approval from an ethics committee for animal experiments?
Author Response
Response to Reviewer 1 Comments
We greatly appreciate the highly constructive comments and tried to address each one of them as detailed below. We hope these changes would satisfactorily address the Reviewer’s suggestions and concerns.
Point 1: I don’t like a title, please consider a title that is more relevant
Response 1: We have changed the title. The new title is: Effect of Danofloxacin Treatment on the Development of Fluoroquinolone Resistance in Campylobacter jejuni in Calves
Point 2: Please try to prepare the introduction in a more concise form with precise aims setting.
Response 2: The introduction is significantly shortened in the revised manuscript, focusing on Campylobacter in cattle and the development of fluoroquinolone(FQ)-resistant C. jejuni. The main aim is indicated in the Introduction (lines 99-102).
Point 3: Please stress the relation between C. jejuni and BRD, this in unclear in present form .
Response 3: This relation is now specfically indicated in the Introduction (lines 54-65), and further emphasized in Results 2.2 section (lines 156-159) in the revised manuscript.
Point 4: Generally, the introduction should be shorter and more clear.
Response 4: The introduction is shortened, with a focus on Campylobacter in cattle and the development of FQ-resistant C. jejuni.
Point 5: Please indicate the main aim of the study.
Response 5: This is indicated in the Introduction (lines 99-102).
Point 6: In the Results part please try to focus on the most relevant results, all results are shown on tables and figures, in my opinion it is not necessary to describe them such.
Response 6: The results section is shortened significantly in the revised manuscript.
Point 7: The part from 612 to 693 is the replication of results, please discuss your results with previous findings and shorten this part to the key results description and discussion.
Response 7: We appreciate the comment and tried to shorten that part in the revised manuscript without losing the context.
Point 8: Please prepare a short conclusion from the study.
Response 8: We have included a short conclusion in the revised manuscript (lines 583-594).
Point 9: I have some doubts about the nomenclature of experimental groups, the terms “control group” and “healthy group” are confusing for me. The term “Control group” suggests animals without treatment, whereas they were inoculated with C. jejuni. Control group actually doesn’t exist in this study. “Healthy group”, in turn, represents calves inoculated with C. jejuni and treated with danofluxacin. Please consider change of these names.
Response 9: We understand how the group naming could be confusing for some readers and thank to the reviewer for this suggestion. We have changed the nomenclature of the groups in the revised manuscript. Control Group is now called Group 1, Healthy Group is called Group 2, and BRD-induced group is called Group 3. Accordingly, we have also modified Table 1 to better depict the different groups of the study.
Point 10: Did the authors receive the approval from an ethics committee for animal experiments?
Response 10: Yes. Information is on lines 617-619.

Reviewer 2 Report
Introduction
The last paragraph must be rewritten. First, the hypothesis of the authors must be recapitulated in two sentences (maximum). Then, the objectives of the study must be clearly stated.
M & M
Table 1 is erroneously placed in 2.1. It should be transferred to 4.1.
Please explain why you did not employ the Glasgow model for respiratory challenge of calves (Gibbs et al.)
Please present in supplementary material all available details regarding all the bacterial isolates used for the challenges.
Results
Table 1, see above.
2.1. The detailed results must be included in a table. As they are now, they are difficult to read and comprehend fully.
2.3. Again, results in a table. Only a brief summary in the text.
2.3. As 2.3.
2.5. Please, the results must be presented in a supplementary material.
2.6. Again, please try to include more details in the tables.
Discussion
This needs to be separated in sub-sections to allow easier reading.
Also, please include a recommendation with a potential address to the regulatory authorities with reference to the results.
Conclusion
It is imperative to include a final section.
Whilst the manuscript presents a lot of seemingly good work, based on a useful idea, it is badly written and assessment could not be made well.
The authors must improve legibility of their manuscript, which at the moment is not well-organised.
In summary, extensive revision and re-evaluation.
Author Response
Response to Reviewer 2 Comments
We thank to the Reviewer for the highly constructive comments and tried to address each one of them as detailed below. We hope these changes would satisfactorily address the Reviewer’s suggestions and concerns.
Point 1: Introduction - The last paragraph must be rewritten. First, the hypothesis of the authors must be recapitulated in two sentences (maximum). Then, the objectives of the study must be clearly stated.
Response 1: The last paragraph of the Introduction has now been revised accordingly, lines 92-102.
Point 2: M& M - Table 1 is erroneously placed in 2.1. It should be transferred to 4.1.
Response 2: Table 1 is in section 2.1 because the journal instructions say that “tables should be placed in the main text near the first time they are cited.” We will leave this up to the Journal’s editorial office.
Point 3: M&M - Please explain why you did not employ the Glasgow model for respiratory challenge of calves (Gibbs et al).
Response 3: This explanation is now included in the revised manuscript (lines 640-647).
Point 4: M&M - Please present in supplementary material all available details regarding all the bacterial isolates used for the challenges.
Response 4: A supplementary material (Table S1) is now attached to the revision. It is also mentioned in the revised manuscript (lines 613 and 773-775).
Point 5: Results - 2.1. The detailed results must be included in a table. As they are now, they are difficult to read and comprehend fully.
Response 5: The entire Results section has been shortened significantly in the revised manuscript. This was also recommended by Reviewer 1. Regarding the request to include the results in a table, even though we greatly appreciate this comment, we believe that it would be very difficult and complicated to include all the results in tables(s) to illustrate the major and some specific findings of the study clearly and concisely. There is ample amount of data generated from several different groups (each having 10 calves) with multiple data types (e.g., colonization levels and percentages by susceptible and resistant isolates from 30 individual calves) over multiple sampling points. We also think that presenting the data in figures instead of tables would better facilitate to visualize the dynamic changes over time.
Point 6: Results - 2.3. Again, results in a table. Only a brief summary in the text.
Response 6: Please see Response 5 above.
Point 7: Results - 2.4. As 2.3.
Response 7: Please see Response 5 above.
Point 8: Results - 2.5. Please, the results must be presented in a supplementary material.
Response 8: We greatly appreciate this suggestion by the reviewer. However, the data (relative percentages of FQ-resistant C. jejuni as determined by the MIC testing; Figure 2b) is a key component of this manuscript (to corroborate the culture results obtained via differential plating, as explained in that section in Results), and thus we believe it should be presented in the main text.
The MIC data on other antibiotics present in the Sensititre panel is not fundamental to the manuscript; therefore, it was further reduced to a short paragraph in this revision. It is included in the main text to briefly mention the antimicrobial susceptibility profiles of C. jejuni from the current study, which we believe may be of interest to the readers.
Point 9: Results - 2.6. Again, please try to include more details in the tables.
Response 9: We very much appreciate this comment from the reviewer. However, we believe that the tables (especially Table 2, and Table 3), as in their current form, already contain a lot of detailed information, including the PFGE genotypes and ciprofloxacin phenotypes and MICs (along with the number of isolates within each category) of the C. jejuni isolates collected throughout the entire study. We have carefully read the relevant section of the manuscript again, but did not find any major data missing from the tables. However, we would gladly consider adding more details to the tables if the reviewer wishes to point those out specifically.
Point 10: Discussion - This needs to be separated in sub-sections to allow easier reading.
Response 10: The journal’s instructions did not specifically ask for separated sub-sections in Discussion, thus we divided the discussion into paragraphs without subsections. As Discussion is shortened in the revision, we believe it is now easier to read now. We will leave this point up to the Journal’s editorial office.
Point 11: Discussion - Also, please include a recommendation with a potential address to the regulatory authorities with reference to the results.
Response 11: We truly appreciate this suggestion. However, we believe that we do not currently have enough data to make a recommendation to the regulatory agencies. As indicated in the last sentence of the Conclusions in the revised manuscript, more studies (including evaluation of various treatment schemes and different FQ antibiotics) are needed to be performed before well-informed conclusions can be drawn.
Point 12: Conclusion - It is imperative to include a final section.
Response 12: We have included a short conclusion in the revised manuscript (lines 583-594).
Point 13: Whilst the manuscript presents a lot of seemingly good work, based on a useful idea, it is badly written and assessment could not be made well. The authors must improve legibility of their manuscript, which at the moment is not well-organized. In summary, extensive revision and re-evaluation.
Response 13: We appreciate the comment and revised the entire manuscript based on the suggestions from both reviewers. We believe the revision has improved the quality and legibility of the manuscript significantly.

Round 2
Reviewer 2 Report
I still maintain that the results are not presented well and that there is room for improvement of the presentation.
As there is disagreement with the authors, I leave this to the academic editor to decide.
Author Response
Response to Reviewer 2 Comments
We greatly appreciate the highly constructive comments and tried to address each one of them as detailed below. We hope these changes would satisfactorily address the Reviewer’s suggestions and concerns.
Point 1: I still maintain that the results are not presented well and that there is room for improvement of the presentation.
Response 1: The results section is shortened significantly in the revised manuscript. We appreciate the comment and revised the entire manuscript based on the suggestions from both reviewers. We believe the revision has improved the quality and legibility of the manuscript significantly.
